

# Interpretation of kinetic isotope fractionation between aqueous Fe(II) and ferrihydrite under a high degree of microbial reduction

**Lei Jiang[1], Chuanjun Wu[1], Mingqing Li[2], Xuegong Li[1], Jiwei Li[1]**

[1]CAS Key Laboratory for Experimental Study under Deep-sea Extreme Environment Conditions, Institute of Deep-sea Science and Engineering, Chinese Academy of Science, Sanya, 572000, China

[2]University of Chinese Academy of Sciences, Beijing 100049, China

**Correspondence to:** Lei Jiang (jl@idsse.ac.cn)

**Abstract.** Microbial dissimilatory iron reduction (DIR) often ceases when the degree of iron mineral reduction is low, at which point isotope fractionation occurs between an aqueous Fe(II) solution and a reactive Fe(III) phase on the surface of ferric (oxyhydro) oxides, forming an equilibrium fractionation factor (~3 ‰). Recent experimental abiotic studies suggest that Fe(II) adsorption onto the mineral surface may affect the isotope fractionation, which reminds us that the isotope exchange may be greatly inhibited during the DIR process. In this study, ferrihydrite is used as a terminal electron acceptor to conduct *Shewanella piezotolerans* WP3 and *Shewanella oneidensis* MR-1 experiments at 0.1 and 15 MPa to ensure a significant variation in the degree of reduction. During the 30-day experiment, the degree of ferrihydrite reduction by *S. piezotolerans* WP3 is 14 % (at 0.1 MPa) and 8 % (at 15 MPa), whereas the degree of ferrihydrite reduction by *S. oneidensis* MR-1 is 39 % (at 0.1 MPa) and 36 % (at 15 MPa). Based on the isotope mass balance, the estimated ranges of iron isotope fractionation for *S. piezotolerans* WP3 and *S. oneidensis* MR-1 are obtained. The former ranges between −3.58 ‰ and −0.88 ‰ (at 0.1 MPa) and between −2.37 ‰ and −0.66 ‰ (at 15 MPa), and the latter ranges between −0.39 ‰ and 0.10 ‰ (at 0.1 MPa) and between −0.6 ‰ and −0.16 ‰ (at 15 MPa). However, it is difficult to distinguish variations in the same bacteria at 0.1 and 15 MPa due to the large estimation ranges of isotope fractionation. In the *S. oneidensis MR-1* experiment, the fractionation factor obtained is significantly different from that obtained in the *S. piezotolerans* WP3 experiment, indicating that kinetic fractionation occurred. In combination with previous studies, we propose a transient modified Fe(II) adsorption mechanism to explain the isotope fractionation between aqueous Fe(II) and ferrihydrite. When the adsorbed Fe(II) exceeds the surface saturation, the atom (isotope) exchange will be suppressed.

## 1 Introduction

Iron is the fourth most abundant element in the Earth's crust and the most common redox-active transition metal (Liu et al., 2001; Liu et al., 2018). This element has been widely used to assess the oxidation state of the environment (Mulholland et al., 2015; Cooper et al., 2017; Ellwood et al., 2019). In an oxygen-restricted reducing environment, iron-reducing microorganisms can effectively use ferric substances as terminal electron acceptors coupled with the oxidation of organic matter or $H_2$ (Li et





al., 2019; Notini et al., 2019). This process of microbial dissimilatory iron reduction (DIR) promotes the reductive dissolution
of iron minerals, forming a wide range of soluble Fe(II), which is easily adsorbed onto the surface of (oxyhydro) oxides and
catalyzes the reduction of contaminants (Newsome et al., 2018). It also controls the cycle of C, N and P (Murray and Hesterberg,
2006; Colombo et al., 2014). Additionally, DIR is probably one of the oldest metabolism processes on Earth (Picard et al.,
2012) and plays an important role in the Precambrian banded iron formation genesis (Percak-Dennett et al., 2011).
Experimental DIR studies, in showing that the partial adsorption of aqueous Fe(II) onto ferric mineral surfaces can inhibit
the degree of microbial reduction (Jaisi et al., 2007). For example, during a DIR experiment lasting ~280 days, the reduction
rates of hematite and goethite were lower than 0.7 % and 4 %, respectively (Crosby et al., 2007). However, this inhibition
seems to no effect on the isotope exchange between aqueous Fe(II) (Fe[II]$_{aq}$) and reactive Fe(III) (Fe[III]$_{reac}$) on the mineral
surface, as the isotope fractionation factor, $\delta^{56}Fe$ (in $^{56}Fe/^{54}Fe$), remains constant at ~−3 ‰ (Wu et al., 2009). This is consistent
with the equilibrium fractionations of abiological experiments (Skulan et al., 2002; Wu et al., 2010). The mechanism of iron
isotope fractionation during the DIR process has been linked with coupled Fe(II)–Fe(III) electron transfer and atom exchange
(ETAE) (Percak-Dennett et al., 2011; Reddy et al., 2015). Recently, A new experimental study reports that an increased Fe(II)
concentration reduced the degree of Fe atom exchange between aqueous Fe(II) and hematite (Frierdich et al., 2015). This
suggests that, when a large amount of aqueous ferrous Fe is produced at a high degree of reduction, the equilibrium of iron
isotope fractionation may not occur.
Ferrihydrite, a less crystalline ferric hydroxide, is found in a wide variety of anoxic environments (Williams and Scherer,
2004). It is highly reactive, so the expected degree of reduction is higher than that of well-crystalline ferric oxides, such as
hematite and goethite, during the DIR process (Li et al., 2012; Poggenburg et al., 2016; Notini et al., 2019; Chanda et al.,
2020). Therefore, in this study, ferrihydrite is employed as a terminal electron acceptor to conduct DIR experiments with
*Shewanella piezotolerans* WP3 and *Shewanella oneidensis* MR-1 at 0.1 and 15 MPa to (i) investigate whether isotopic
fractionation equilibrium occurs at a high degree of reduction and (ii) analyze the possible causes of kinetic isotope
fractionation.

## 2 Materials and Methods

### 2.1 Ferrihydrite substrate

Ferrihydrite solids used in this study were synthesized in a laboratory by slowly neutralizing ferric nitrate with potassium
hydroxide to a pH of 7.5, and then drying it for 36 h in a freeze drier. The particles obtained were deformed, with approximate
dimensions of $0.3 \times 0.8$ μm, determined by SEM. The ferrihydrite powders were partially dissolved using dilute HCl at
different time intervals, and the isotope compositions of the dissolved and remaining undissolved ferrihydrite components
were measured, which indicated that the ferrihydrite was isotopically homogenous. Large samples were completely dissolved
with 0.5 M HCl, indicating that the initial ferrihydrite powder isotope composition was $0.10 \pm 0.06$ ‰.



## 2.2 Bacterial strains and culture mediums


The dissimilatory Fe(III)-reducing strain, *S. piezotolerans* WP3, was purchased from Shanghai Jiao Tong University and grown
at 20°C in 2216E Marine Medium with constant shaking at 150 rpm. *S. oneidensis* MR-1 was purchased from the American
Type Culture Collection and grown at 25°C in lysogeny broth (LB) with constant shaking at 150 rpm. All experiments were
performed in an anoxic chamber at room temperature. At the beginning of each experiment, cells were harvested and washed
twice, and obtained a final concentration of approximately $10^6$ cells ml$^{-1}$. The 2216E Marine Medium contained 5 g tryptone,
1 g yeast extract, and 34 g NaCl in one liter of water, as well as 10 g tryptone, 5 g yeast extract, and 10 g NaCl in one liter of
water in LB culture. The pH of the mediums was adjusted to 7.0 by neutralization with 1 M KOH, then they were added to 50
ml serum bottles and sterilized at 120°C for 20 min. The serum bottles were capped with rubber stoppers and flushed with $N_2$
to exclude $O_2$ until the concentration was lower than 2 μmol/L (detected by an oxygen probe manufactured by Unisense,
Denmark). Subsequently, the mediums used for growth and DIR experiments were added to 50 ml sterile plastic syringes,
followed by addition of 1 g ferrihydrite and bacteria. The syringes were then sealed with PE material stoppers and placed in
steel adjustable pressure vessels.

## 2.3 Sampling and extraction procedures


Experiments were performed using *S. piezotolerans* WP3 and *S. oneidensis* MR-1 at 0.1 and 15 MPa to determine the reduction
rate, degree of reduction, and possible isotope fractionation of iron. Iron species were harvested for concentration and isotope
composition analysis at 2, 5, 20, and 30 days. After centrifugation, the supernatant was extracted and filtrated using a 0.2-μm
filter and then HCl was added to a 0.5 M final concentration. The remaining solid component was then digested with 0.1 M
HCl for 15 min, which removed the majority of the sorbed Fe(II) (Fe[II]$_{sorb}$) (Percak-Dennett et al., 2011) and a small amount
of ferric substrate, determined using Fe concentration measurements. After extraction with 0.1 M HCl, the remaining fractions
were extracted using 15 ml 0.5 M HCl until completely dissolved. During these time intervals, the concentration of Fe(II) and
total Fe was analyzed by the ferrozine method (Stookey, 1970), and the Fe(III) concentration was calculated from the difference.
Concentration errors of Fe(II) and total Fe were calculated using standard deviation of repeat measurements, and Fe(III)
concentration errors were determined using the square root of the sum of squared Fe(II) and total Fe concentration errors. The
results are shown in Table 1.

## 2.4 Iron isotope measurements


The measurement of iron isotopes was performed in the Key Laboratory of Crust-Mantle Materials and Environments,
University of Science and Technology of China, Chinese Academy of Sciences. All Fe-containing solutions, including samples
of aqueous Fe(II) and 0.1 M HCl extracts, were purified using anion-exchange chromatography before the iron isotope
measurements were analyzed using a multi-collector inductively coupled plasma mass spectrometer. Detailed experimental





procedures have previously been reported by Huang et al. (2011). All the isotopic compositions were expressed as $\delta^{56}$Fe values
relative to the iron reference material IRMM-014, as follows:
$$\delta^{56}\text{Fe}(‰) = [\frac{(^{56}\text{Fe}/^{54}\text{Fe})_{sample}}{(^{56}\text{Fe}/^{54}\text{Fe})_{IRMM-014}} -1] \times 10^3 \qquad (1)$$
The iron isotopic fractionation between phases A and B is defined as:
$$\Delta^{56}\text{Fe}_{A-B} = \delta^{56}\text{Fe}_A - \delta^{56}\text{Fe}_B \qquad (2)$$
The external precision of the measured $\delta^{56}$Fe values was better than ±0.05 ‰ (1δ), based on long-term repeated analyses. The
analysis results are listed in Table 2.
**3 Results**
**3.1 Ferrihydrite reduction**
The ferrous Fe content varied significantly during the course of the bioreduction experiments. The total Fe(II) concentration
for the initial *S. pizotolerans* WP3 and *S. oneidensis* MR-1 experiments increased rapidly under both pressures. Within five
days, the total Fe(II) concentration in the *S. pizotolerans* WP3 experiment reached 3.03 mM L$^{-1}$ (at 0.1 MPa) and 2.00 mM
L$^{-1}$ (at 15 MPa). In the *S. oneidensis* MR-1 experiment, the total Fe(II) concentration reached 12.16 mM L$^{-1}$ (at 0.1 MPa) and
10.14 mM L$^{-1}$ (at 15 MPa). These results indicate that the initial rate of ferrihydrite reduction by *S. oneidensis* MR-1 was
significantly higher than it was by *S. pizotolerans* WP3 under the same pressure, and it was also slightly faster at 0.1 than at
15 MPa for the same strain (Fig. 1; Table 1). As the reduction progressed, the rate decreased so sharply that the concentrations
of produced Fe(II) remained almost constant over the second half of the experiments. *S. oneidensis* MR-1 cultures reduced
ferrihydrite by 39 % (at 0.1 MPa) and 36 % (at 15 MPa), while *S. pizotolerans* WP3 reduced ferrihydrite slightly less (by 14 %
and 8 %, respectively), at the end of the experiments. The degree of reduction obtained in the experiments was significantly
higher than in previous studies using well-crystalline ferric oxides (Beard et al., 2010).
The aqueous Fe(II) was measured by iron in solution because aqueous Fe(II) was the only iron phase in the aqueous fraction,
yet the sorbed Fe(II) was determined by the sum of Fe(II) removed in the 0.1 and 0.5 M HCl extracts. The initial concentration
ratios of Fe(II)$_{sorb}$ and Fe(II)$_{aq}$ for *S. pizotolerans* WP3 on day two were 10.8 (at 0.1 MPa) and 29.5 (at 15 MPa), and for *S.*
*oneidensis* MR-1 they were 5.4 (at 0.1 MPa) and 4.4 (at 15 MPa). Finally, the concentration ratios decreased to 3.2 (at 0.1
MPa) and 2.6 (at 15 MPa) for *S. pizotolerans* WP3, and 5.4 (at 0.1 MPa) and 2.2 (at 15 MPa) for *S. oneidensis* MR-1 (Table
1), indicating that the available sorption sites on ferrihydrite surface decreased with the accumulation of Fe(II) produced before
the surface site capacity reached saturation.



### 3.2 Iron isotope compositions

Fe isotope compositions are shown in Table 2. The isotope composition values of aqueous Fe(II) produced using *S. pizitolerans* WP3 varied slightly at 0.1 MPa (average: ~−1.5 ‰), whereas the isotopic compositions for 0.1 M HCl extracts increased constantly with time, which changed from ~−0.8 ‰ on day two to ~−0.3 ‰ on day 30 (Fig. 2a). Under high pressure experiments (15 MPa), $\delta^{56}Fe(II)_{aq}$ and $\delta^{56}Fe(II)_{0.1\,M\,HCl}$ changed with no obvious trends, and the average values were ~−1.5 ‰ and ~−1.0 ‰, respectively (Fig, 2b). In the *S. oneidensis* MR-1 experiment, however, $\delta^{56}Fe(II)_{aq}$ and $\delta^{56}Fe(II)_{0.1\,M\,HCl}$ increased slightly in the same manner, and there were no significant differences between the results at 0.1 and 15 MPa (Fig. 2c, d). Comparing these two bacterial experiments, the isotopic composition of aqueous Fe(II) and 0.1 M HCl extracts were significantly different, which may be due to the relative different amounts of Fe species, including $Fe(II)_{aq}$, $Fe(II)_{sorb}$ and $Fe(III)_{reac}$, and isotope fractionation factors (Crosby et al., 2007).

## 4 Discussion

### 4.1 Sorbed Fe(II) suppression of Fe(III) bioreduction

Initially, Fe(II) produced by *S. oneidensis* MR-1 was almost three times the amount produced by *S. pizotolerans* WP3 under the same pressure, and the reduction rate for the same bacteria at 0.1 MPa was slightly higher than at 15 MPa. The results indicate that the species of bacteria and the amount of pressure can both significantly influence the rate of DIR. The mechanism may be related to the enzyme activity of the strain, which can be affected by pressure, according to Picard et al. (2012) and Wu et al. (2009). As the reduction process progressed, the rate had decreased significantly by day five (*S. pizotolerans* WP3) and day 10 (*S. oneidensis* MR-1), whereas the living cell concentration and pH in the reduction reactor remained constant, thus ruling out the effect of biomass and pH on the DIR rate. Many other factors can influence the DIR rate, such as the crystallinity of ferric minerals (Picard et al., 2012 and references therein), electron shuttles (MacDonald et al., 2011), and the presence of reduced graphene (Liu et al., 2018). Experimental studies by Jaisi et al. (2007) found that the adsorption of Fe(II) on the surface of old cells partly resulted in the cessation of bioreduction activity. Additionally, when a certain amount of ferrous iron was presorbed onto clay minerals, the reduction rate and degree of reduction continued to decrease with the increasing presorbed Fe(II). Therefore, the most likely cause for the rate decrease in this experiment is the Fe(II) adsorption onto the cell and ferrihydrite surfaces, and the mechanism of forming an Fe(II)-bearing layer (Hansel et al., 2004) or altering the surface potential (Roden and Urrutia, 2002).

### 4.2 Estimation of the range of isotope fractionation

A number of laboratory studies have shown that the underlying mechanism of Fe isotope fractionation during DIR is linked with the coupled Fe(II)–Fe(III) electron transfer and atom exchange on the surface of (oxyhydro) oxides (Tangalos et al., 2010). A isotope fractionation factor of −2.95 ± 0.19 ‰ between $Fe(II)_{aq}$ and $Fe(III)_{reac}$ was obtained during hematite bioreduction,





which was consistent with the abiotic equilibrium fractionation of −3.1 ‰ at 22 °C determined by Skulan et al. (2002), and
was also identical to the Fe(II)$_{aq}$–Fe(III)$_{reac}$ isotope fractionation factor of −2.87 ± 0.19 ‰ in abiological hematite reduction
experiments (Wu et al., 2010). Previous studies have shown that biogenic Fe(II) produced during DIR using ferrihydrite was
enriched with light Fe isotopes (Tangalos et al., 2010); however, it was difficult to determine the isotopic compositions of
Fe(II)$_{sorb}$ and Fe(III)$_{reac}$ because ferrihydrite was not amenable to partial acid extraction (Percak-Dennett et al., 2011).
Additionally, the possible formation of secondary minerals, such as magnetite, added complexity to the interpretation of Fe
isotopic composition (Reddy et al., 2015).
In this study, the isotopic composition of the 0.1 M HCl extract was a mixture of Fe(II)$_{sorb}$ and Fe(III)$_{reac}$, making it difficult
to use the methods of Wu et al. (2009) to calculate the exact $\delta^{56}$Fe(II)$_{sorb}$ and $\delta^{56}$Fe(III)$_{reac}$ for *S. pizotolerans* WP3 and *S.*
*oneidensis* MR-1 at 0.1 and 15 MPa, respectively. Based on the isotope mass balance, $\delta^{56}$Fe value of the 0.1 M HCl extract
can be calculated using the equation:
$$\delta^{56}\text{Fe}_{0.1\text{ M HCl}} = X^{\text{HCl}}_{\text{Fe(II)}_{sorb}} \delta^{56}\text{Fe(II)}_{sorb} + X^{\text{HCl}}_{\text{Fe(III)}_{reac}} \delta^{56}\text{Fe(III)}_{reac} \tag{3}$$
where $X$ is the mole fraction.
Previous experimental studies suggested that the adsorption of Fe(II) onto mineral surfaces makes the aqueous Fe(II) pool
depleted in heavy Fe isotope (Rouxel et al., 2008). For example, Icopini et al. (2004) pointed out that the isotopic composition
of sorbed Fe(II) pool was ~2.7 ‰–3.7 ‰ heavier than aqueous Fe(II). Beard et al. (2010) indicated that the equilibrium
fractionation factor between sorbed Fe(II) and aqueous Fe(II) was 1.24 ± 0.14 ‰ during microbial DIR of goethite. In contrast,
Crosby et al. (2007) obtained the isotope fractionation factor of 0.3 ‰ in hematite reduction, and 0.87 ‰ in goethite reduction.
If it is assumed that the isotopic composition of sorbed Fe(II) is equal to aqueous Fe(II), we can obtain an upper-limit isotope
Fe(III)$_{reac}$ composition, according to equation (3), where $\delta^{56}$Fe$_{0.1\text{ M HCl}}$, $X^{\text{HCl}}_{\text{Fe(II)}_{sorb}}$ and $X^{\text{HCl}}_{\text{Fe(III)}_{reac}}$ are known. In turn, assuming that
the isotopic composition of sorbed Fe(II) is identical to that of the 0.1 M HCl extract, the lower limit of isotopic composition
is given. Therefore, we have determined the approximation range of Fe(II)$_{aq}$–Fe(III)$_{reac}$ isotope fractionation. In Table 3, the
average estimated maximum and minimum $\Delta^{56}$Fe$_{\text{Fe(II)aq-Fe(III)reac}}$ fractionations produced by *S. pizitolerans* WP3 at 0.1 MPa
were, respectively, ~−3.58 ‰ and −0.88 ‰, and they were ~−2.37 ‰ and ~−0.66 ‰, respectively, at 15 MPa. The results
covered or approached the isotope fractionation of ~−3 ‰ obtained by predecessors (Crosby et al., 2007), so it was not clear
whether the equilibrium isotope fractionation was reached. However, in the *S. oneidensis* MR-1 experiment, the average
maximum isotope fractionations were ~−0.39 ‰ at 0.1 MPa and ~−0.60 ‰ at 15 MPa, which were significantly less than the
average minimum of *S. pizitolerans* WP3 at both pressures (Fig. 3), indicating that kinetic isotope fractionation had occurred.
**4.3 Interpretation for kinetic isotope fractionation**
Several experiments have suggested that there are iron isotope fractionations between aqueous Fe(II), sorbed Fe(II), and active
Fe(III) on the (oxyhydro) oxide surfaces during microbial DIR. The mechanism involves the reductive dissolution of ferric
atoms on the surface of (oxyhydro) oxides, followed by adsorption of aqueous Fe(II) on mineral surfaces and ETAE process





between sorbed Fe(II) and reactive ferric atoms (Shi et al., 2016). Surface defects resulting in the local charge imbalance have
been shown to increase the driving force of ETAE (Notini et al., 2019). However, in recent years, some studies have reported
that an atom exchange can occur between aqueous Fe(II) and the bulk structural ferric Fe (Handler et al., 2014; Frierdich et
al., 2015). The "redox-driven conveyor belt" model has been proposed to explain this. The model involves the conduction of
electrons from sorption sites to dissolution sites through bulk crystal, resulting in the reductive dissolution of Fe atoms at
separate surface sites, thereby generating the aqueous Fe(II) again (Neumann et al., 2015). This model is consistent with the
absence of secondary minerals formation and changes in particle shape and size at Fe atoms exchange between hematite or
goethite and aqueous Fe(II) (Handler et al., 2014).
The latest research shows that the amount of Fe atom exchange between aqueous Fe(II) and ferric oxides increases with the
increasing amount of sorbed Fe(II); however, a lower amount of Fe atom exchange occurs when sorbed Fe(II) exceeds the
surface site capacity (Frierdich et al., 2015). One possible explanation for this is that Fe atom exchange at a lower coverage of
sorbed Fe(II) on the surface of ferric oxides is mainly controlled by interfacial electron transfer that is driven by local charge
imbalance at structural distinct surface sites. As the coverage of sorbed Fe(II) on the surface of ferric oxides increases, the
propensity of interfacial electron transfer potency diminishes (Frierdich et al., 2015). However, Handler et al. (2014) found
that there was no net sorbed Fe(II) on the hematite surface, and some isotope exchanges were still observed, indicating that
isotope exchange between aqueous Fe(II) and structural Fe(III) on the hematite surface may be carried out by transient
adsorption, which was suppressed when the surface site capacity reached saturation. Over the course of 30 days, the degree of
ferrihydrite reduction was less than 14 % in the *S. pizotolerans* WP3 experiment, and the surface position did not exceed the
surface saturation, thus forming a larger isotope fractionation (Fig. 4a). However, a larger degree of ferrihydrite reduction
(18 %~39 %) was obtained in the *S. oneidensis* MR-1 experiment, where the amount of Fe(II) adsorption exceeded the surface
saturation, so the charge-neutral configurations could be reconstructed. This resulted in the suppression of the interfacial
electron transfer and the transient adsorption of Fe(II) (Fig. 4b).
**5. Conclusions**
The results show that isotope equilibrium fractionation occurs due to the rapid ETAE process between aqueous Fe(II) and
Fe(III) on the surface of (oxyhydro) oxides at a low degree of reduction during DIR, where the amount of sorbed Fe(II) is
below the surface site capacity. However, at a high degree of reduction, the amount of stable adsorbed Fe(II) on mineral surface
increases with the increasing aqueous Fe(II), and the driving force for ETAE process decreases in response to the healing of
surface structure imperfections. At the same time, the isotope exchange between aqueous Fe(II) and Fe(III) on mineral surface
via the transient adsorption of Fe(II) diminishes with a decrease in surface sites. When the surface site is saturated with
adsorbed Fe(II), the atom exchange is significantly suppressed, thus exhibiting the characteristics of kinetic isotope
fractionation.



*Data availability. All data used are listed in the article.*
*Author contributions.* LJ, XL and JL designed the experiments. ML conducted the experiments and prepared the original draft.
LJ and CW contributed to discussion and interpretation of the data and the writing of the paper.
*Competing interests*. The authors declare that they have no conflict of interest.
*Acknowledgements*. Special thanks go to Xiang Xiao (Shanghai Jiao Tong University), who provided indispensable *Shewanella*
*piezotolerans* strain WP3. We thank Juezhi Lin, Yang Xin and Jiyue Sun for excellent laboratory support. We also would like
to acknowledge Faboya Lekan (Guangzhou institute of Geochemistry, CAS) for his help on improving the manuscript.
*Financial support.* This research has been supported by the National Natural Science Foundation of China (grant nos.
41403050 and 41873068).

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



**Table 1.** Fe concentrations in aqueous 0.1 and 0.5 M HCl fractions, degrees of reduction, and ratios of $M_{Fe(II)sorb}/M_{Fe(II)aq}$ for *Shewanella*
*piezotolerans* WP3 and *Shewanella oneidensis* MR-1 experiments at 0.1 and 15 MPa

| Pressure (MPa) | Day | Aqueous (mM) | | 0.1 M HCl (mM) | | | | 0.5 M HCl (mM) | | | | Reduction extent (%) | $M_{Fe(II)sorb}/M_{Fe(II)aq}$ |
|---|---|---|---|---|---|---|---|---|---|---|---|---|---|
| | | Fe(II) | error | Fe(II) | error | Fe(total) | error | Fe(II) | error | Fe(total) | error | | |
| *Shewanella piezotolerans* WP3 | | | | | | | | | | | | | |
| | 2 | 0.24 | 0.01 | 1.10 | 0.04 | 1.50 | 0.03 | 1.48 | 0.74 | 44.57 | 1.91 | 0.06 | 10.8 |
| | 5 | 0.36 | 0.05 | 1.68 | 0.02 | 2.13 | 0.01 | 0.99 | 0.05 | 41.95 | 0.27 | 0.07 | 7.4 |
| 0.1 | 10 | 0.51 | 0.04 | 2.05 | 0.16 | 2.71 | 0.20 | 1.06 | 0.02 | 41.08 | 0.80 | 0.08 | 6.1 |
| | 20 | 1.13 | 0.02 | 3.60 | 0.01 | 4.75 | 0.06 | 1.13 | 0.05 | 40.48 | 0.05 | 0.13 | 4.2 |
| | 30 | 1.08 | 0.22 | 2.59 | 0.12 | 3.62 | 0.04 | 0.86 | 0.06 | 27.39 | 5.19 | 0.14 | 3.2 |
| | 2 | 0.04 | 0.00 | 0.14 | 0.02 | 0.33 | 0.04 | 0.95 | 0.15 | 43.66 | 1.91 | 0.03 | 29.5 |
| | 5 | 0.35 | 0.10 | 0.66 | 0.32 | 0.85 | 0.35 | 0.99 | 0.03 | 41.92 | 0.08 | 0.05 | 4.7 |
| 15 | 10 | 0.75 | 0.09 | 1.22 | 0.41 | 1.60 | 0.53 | 1.07 | 0.02 | 40.82 | 0.29 | 0.07 | 3.0 |
| | 20 | 0.63 | 0.02 | 0.96 | 0.07 | 1.24 | 0.05 | 1.11 | 0.08 | 42.47 | 0.96 | 0.06 | 3.3 |
| | 30 | 0.79 | 0.14 | 1.04 | 0.00 | 1.34 | 0.01 | 0.98 | 0.05 | 33.93 | 3.18 | 0.08 | 2.6 |
| *Shewanella oneidensis* MR-1 | | | | | | | | | | | | | |
| | 2 | 1.31 | 0.02 | 6.43 | 0.12 | 8.61 | 0.14 | 0.70 | 0.03 | 37.21 | 0.35 | 0.18 | 5.4 |
| | 5 | 2.10 | 0.05 | 9.26 | 0.23 | 12.86 | 0.05 | 0.80 | 0.03 | 32.25 | 0.48 | 0.26 | 4.8 |
| 0.1 | 10 | 3.49 | 0.15 | 10.98 | 0.72 | 15.13 | 0.15 | 0.71 | 0.05 | 26.46 | 0.16 | 0.34 | 3.4 |
| | 20 | 4.11 | 0.14 | 10.20 | 0.07 | 15.80 | 0.53 | 0.64 | 0.03 | 24.38 | 1.22 | 0.34 | 2.6 |
| | 30 | 4.59 | 0.04 | 9.86 | 0.46 | 13.67 | 0.70 | 0.64 | 0.03 | 20.32 | 0.13 | 0.39 | 2.3 |
| | 2 | 1.07 | 0.00 | 3.73 | 0.37 | 4.76 | 0.51 | 1.02 | 0.00 | 52.98 | 2.63 | 0.10 | 4.4 |
| | 5 | 1.69 | 0.15 | 7.40 | 0.23 | 9.56 | 0.40 | 1.05 | 0.11 | 46.22 | 1.02 | 0.18 | 5.0 |
| 15 | 10 | 2.45 | 0.17 | 7.47 | 0.48 | 10.99 | 1.71 | 0.94 | 0.04 | 31.44 | 0.27 | 0.24 | 3.4 |
| | 20 | 3.01 | 0.15 | 9.25 | 0.20 | 12.71 | 0.14 | 1.07 | 0.03 | 38.76 | 1.12 | 0.24 | 3.4 |
| | 30 | 4.21 | 0.28 | 8.43 | 0.58 | 11.22 | 1.40 | 0.70 | 0.03 | 21.78 | 1.22 | 0.36 | 2.2 |






**Table 2.** Fe isotope compositions of aqueous and 0.1 M HCl extract for *Shewanella piezotolerans* WP3 and *Shewanella oneidensis* MR-1
experiments at 0.1 and 15 MPa

| Pressure (MPa) | Day | $\delta^{56}Fe_{aq}$ (‰) | 2sd | $\delta^{56}Fe_{0.1\ M\ HCl}$ (‰) | 2sd |
|---|---|---|---|---|---|
| *Shewanella piezotolerans* WP3 | | | | | |
| | 2 | -1.53 | 0.00 | -0.81 | 0.03 |
| | 5 | -1.62 | 0.02 | -0.82 | 0.06 |
| 0.1 | 10 | -1.64 | 0.02 | -0.69 | 0.05 |
| | 20 | -1.34 | 0.05 | -0.39 | 0.02 |
| | 30 | -1.29 | 0.05 | -0.28 | 0.02 |
| | 2 | -1.74 | 0.03 | -0.73 | 0.02 |
| | 5 | -1.50 | 0.05 | -0.99 | 0.03 |
| 15 | 10 | -1.32 | 0.00 | -0.87 | 0.05 |
| | 20 | -1.73 | 0.02 | -0.98 | 0.06 |
| | 30 | -1.44 | 0.02 | -0.87 | 0.03 |
| *Shewanella oneidensis* MR-1 | | | | | |
| | 2 | -0.47 | 0.05 | -0.31 | 0.04 |
| | 5 | -0.43 | 0.04 | -0.20 | 0.03 |
| 0.1 | 10 | -0.25 | 0.02 | -0.17 | 0.05 |
| | 20 | -0.15 | 0.03 | -0.12 | 0.02 |
| | 30 | -0.14 | 0.04 | -0.13 | 0.03 |
| | 2 | -0.56 | 0.03 | -0.48 | 0.04 |
| | 5 | -0.51 | 0.03 | -0.25 | 0.02 |
| 15 | 10 | -0.46 | 0.02 | -0.20 | 0.03 |
| | 20 | -0.30 | 0.05 | -0.20 | 0.04 |
| | 30 | -0.24 | 0.03 | -0.16 | 0.00 |






**Table 3.** The estimated ranges of iron isotope fractionation between aqueous Fe(II) (Fe[II]$_{aq}$) and reactive Fe(III) (Fe[III]$_{reac}$) on the surface
of ferrihydrite for *S. piezotolerans* WP3 and *S. oneidensis* MR-1—when assuming that $\delta^{56}$Fe(II)$_{sorb}$ is equal to $\delta^{56}$Fe(II)$_{aq}$, the maximum
$\Delta^{56}$Fe$_{Fe(II)aq-Fe(III)reac}$ is obtained from the difference between $\delta^{56}$Fe(II)$_{aq}$ and $\delta^{56}$Fe(III)$_{reac}$(Max); likewise, the minimum $\Delta^{56}$Fe$_{Fe(II)aq-Fe(III)reac}$
can be calculated assuming that $\delta^{56}$Fe(II)$_{reac}$ is equal to $\delta^{56}$Fe(II)$_{0.1\ M\ HCl}$

| Pressure (MPa) | Day | $X^{0.1\ M\ HCl}_{Fe(II)sorb}$ | $X^{0.1\ M\ HCl}_{Fe(III)reac}$ | $\delta^{56}$Fe(III)$_{reac}$ (Max) | $\delta^{56}$Fe(III)$_{reac}$ (Min) | $\Delta^{56}$Fe$_{Fe(II)aq-Fe(III)reac}$ (Max) | $\Delta^{56}$Fe$_{Fe(II)aq-Fe(III)reac}$ (Min) |
|---|---|---|---|---|---|---|---|
| *Shewanella piezotolerans* WP3 | | | | | | | |
| | 2 | 0.73 | 0.27 | 1.17 | -0.81 | -2.70 | -0.72 |
| | 5 | 0.79 | 0.21 | 2.18 | -0.82 | -3.80 | -0.79 |
| 0.1 | 10 | 0.76 | 0.24 | 2.25 | -0.69 | -3.89 | -0.95 |
| | 20 | 0.76 | 0.24 | 2.60 | -0.39 | -3.94 | -0.95 |
| | 30 | 0.71 | 0.29 | 2.26 | -0.28 | -3.56 | -1.01 |
| | 2 | 0.42 | 0.58 | 0.01 | -0.73 | -1.74 | -1.01 |
| | 5 | 0.77 | 0.23 | 0.74 | -0.99 | -2.24 | -0.51 |
| 15 | 10 | 0.76 | 0.24 | 0.58 | -0.87 | -1.90 | -0.45 |
| | 20 | 0.78 | 0.22 | 1.64 | -0.98 | -3.37 | -0.75 |
| | 30 | 0.78 | 0.22 | 1.18 | -0.87 | -2.62 | -0.58 |
| *Shewanella oneidensis* MR-1 | | | | | | | |
| | 2 | 0.75 | 0.25 | 0.18 | -0.31 | -0.65 | -0.16 |
| | 5 | 0.72 | 0.28 | 0.40 | -0.20 | -0.83 | -0.23 |
| 0.1 | 10 | 0.73 | 0.27 | 0.06 | -0.17 | -0.32 | -0.09 |
| | 20 | 0.65 | 0.35 | -0.06 | -0.12 | -0.10 | -0.03 |
| | 30 | 0.72 | 0.28 | -0.11 | -0.13 | -0.03 | -0.01 |
| | 2 | 0.78 | 0.22 | -0.21 | -0.48 | -0.35 | -0.08 |
| | 5 | 0.77 | 0.23 | 0.65 | -0.25 | -1.16 | -0.26 |
| 15 | 10 | 0.68 | 0.32 | 0.34 | -0.20 | -0.80 | -0.26 |
| | 20 | 0.73 | 0.27 | 0.08 | -0.20 | -0.37 | -0.10 |
| | 30 | 0.75 | 0.25 | 0.08 | -0.16 | -0.32 | -0.08 |





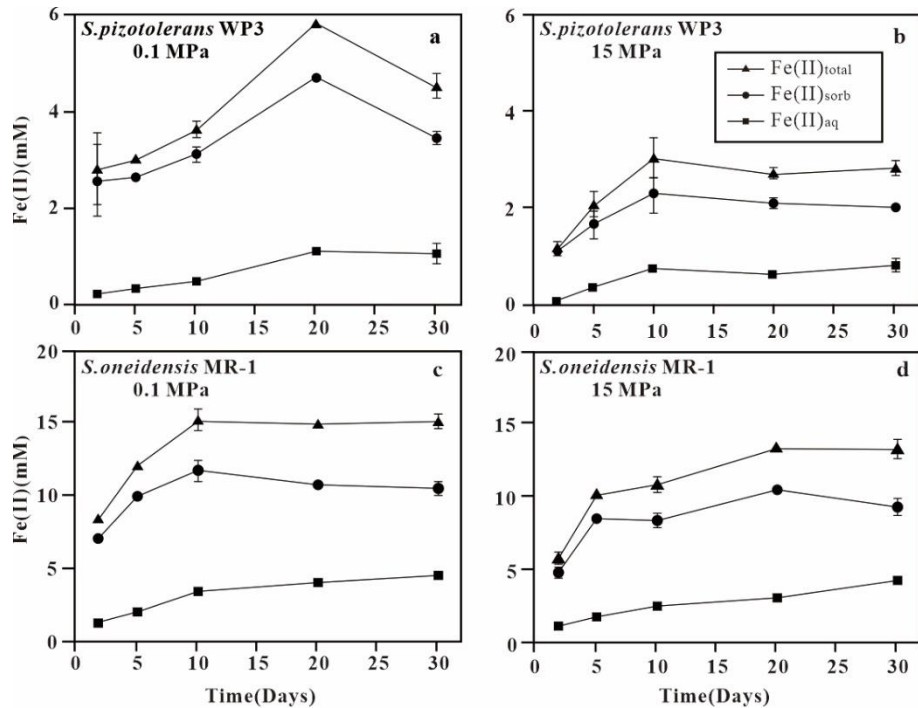


**Figure 1.** Temporal variations of aqueous Fe(II), adsorbed Fe(II), and total Fe(II) concentrations for ferrihydrite reduction using *Shewanella*

*piezotolerans* WP3 and *Shewanella oneidensis* MR-1 at 0.1 and 15 MPa, respectively.






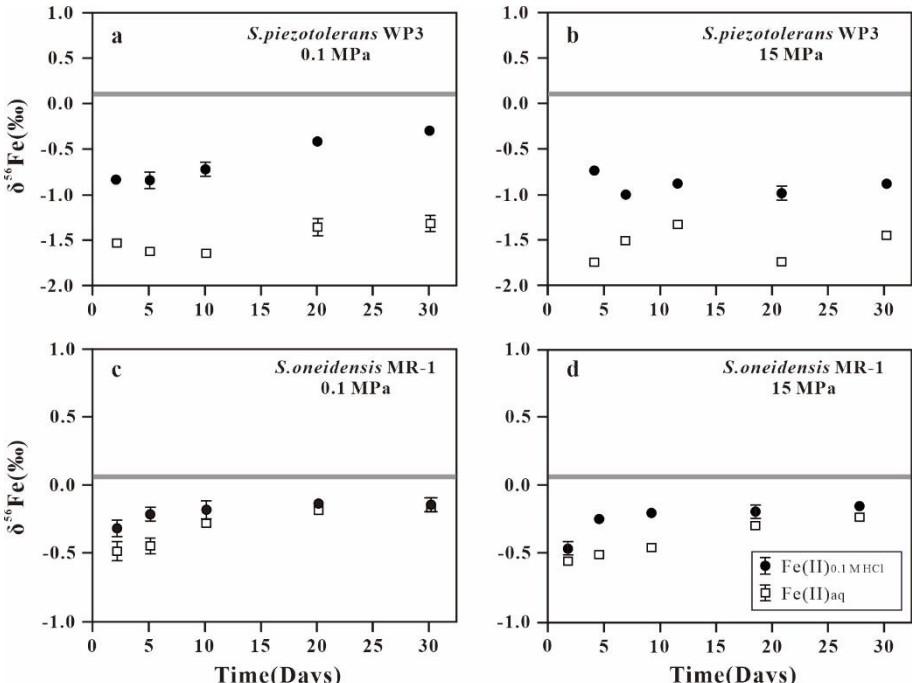

**Figure 2.** Fe isotope compositions of aqueous Fe(II) and 0.1 M HCl extract as a function of time for ferrihydrite reduction using *Shewanella piezotolerans* WP3 and *Shewanella oneidensis* MR-1 at 0.1 and 15 MPa





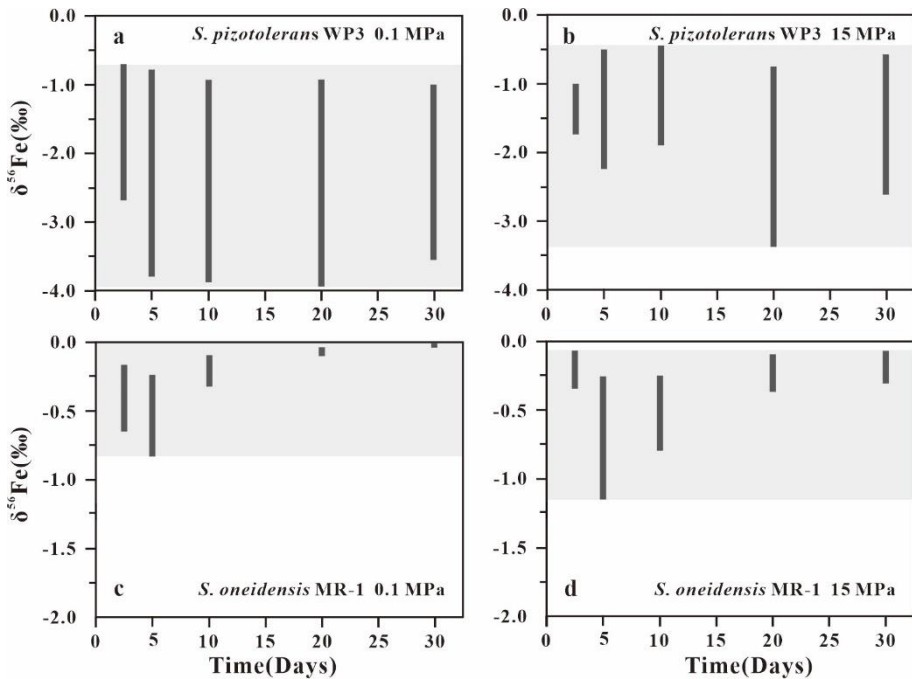

**Figure 3.** Temporal variations in the range of isotope fractionation factors for *Shewanella piezotolerans* WP3 and *Shewanella oneidensis* MR-1 reduction of ferrihydrite at 0.1 and 15 MPa





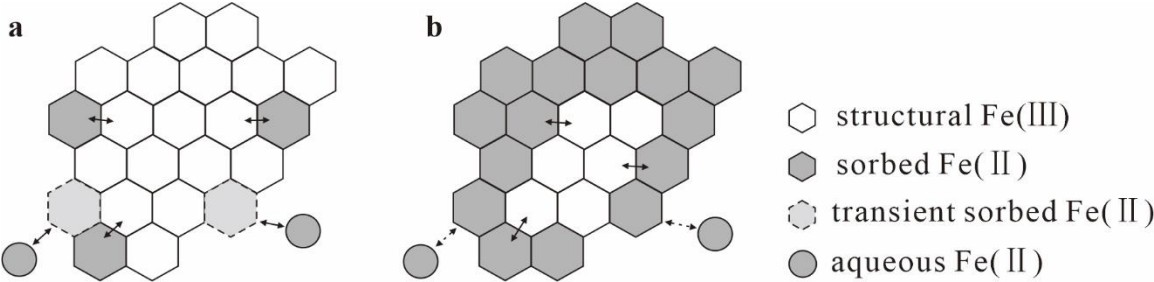

**Figure 4.** Comparison of isotope fractionation mechanisms under low and high degrees of bacterial-mediated reduction of ferrihydrite—**(a)** the amount of Fe(II) adsorbed onto ferrihydrite surface is finite at the low degree of reduction; therefore, the aqueous Fe(II) and surface structural ferric layer isotope exchange is free, and the equilibrium isotope fractionation can occur; **(b)** however, at the high degree of reduction, the outermost layer of ferrihydrite is saturated with sorbed Fe(II) that blocks isotope exchange via short-term revised adsorption between aqueous Fe(II) and surface structural ferric layer, resulting in kinetic isotope fractionation.