# Peer review of "Interpretation of kinetic isotope fractionation between aqueous Fe(II) and ferrihydrite under a high degree of microbial reduction"

_Biogeosciences, 2020_

## Referee Comment (RC1) · Anonymous Referee #1 · 21 Jun 2020

The authors aimed at determining the isotope fractionation factor of iron during the reductive dissolution of ferrihydrite by two different bacterial strains at two different pressures. The authors hypothesize that isotopic fractionation between Fe(II) and Fe(III) does not reach equilibrium when the sorption capacity of ferrihydrite for Fe(II) is reached and isotopic exchange becomes kinetically hindered. By this, the authors intend to confirm conclusions from Frierdrich et al. (2015) in the context of microbial iron reduction. Confirming these conclusions in a different context is valid but not highly innovative. Furthermore, Fe fractionation upon microbial reduction has been extensively investigated (see the multiple studies involving B.L. Beard, C.M. Johnston or E. Roden) so that is hard to identify knowledge gaps. In any case, the calculated fractionation fac-

tors tend to increase over time in experiments with the largest extent of iron reduction. The authors compiled a nice data set. However, in my opinion, the data are not suitable to support a scientific publication due to the limitations to interpret them. This is the predominant reason why I propose to reject the manuscript. My concerns about the limitations are elaborated in the following. The authors use ferrihydrite in their experiments. As the authors realize, ferrihydrite readily transforms into secondary minerals in the presence of Fe(II) depending on, among other factors, Fe(II) concentration. Hence, it can be assumed that different types of secondary iron minerals have been formed in the experiments depending on the rates and extent of Fe(III) reduction. Changes in mineralogy, obviously, effect isotope fractionation and without quantitative information of the Fe isotope signature of the various Fe species it is very difficult to interpret fractionation factor. I also have several other concerns about the interpretation of the data: According to the methodology about 1 g Fh were added to 50 mL medium. This should yield a Fe concentration of about 120 mM. This implies that only around 50 % of total Fe was recovered, which questions the isotope values for Fe(III) when the digestion was not quantitative. The trend that Fe recovery decreases with progressing reaction might reflect Fe mineral transformation (e.g. magnetite formation). The ratio Fe(II)(0.1M HCl) /( Fetot(0.1M HCl) +Fetot(0.5M HCl)) exceeds 0.25, which is larger than a realistic concentration of surface sites (about 0.2 per Fe for HFO). This implies that not all extracted Fe(II) is adsorbed Fe but includes structurally bound Fe(II). The authors do not mention anything about pH. Does the pH change throughout the reaction (no buffer is present in the medium) and how would pH effect fractionation. Considering these uncertainties, I am sceptic that the data set could be used to rigorously discussing fractionation mechanisms or deriving reliable fractionation factors. I have also a couple of minor comments: Why did the authors vary the pressure? The experimental design is not justified. Varying the reduction rates or manipulating the Fe(II) / Fe(tot) ratios could have been easier achieved by adapting the bacteria / Fh ratio. Using different organisms and pressures creates unnecessary ambiguity. Fh is produced by neutralizing a Fe(III) NO3 solution with KOH. The authors do not mention any purification step before

freeze drying, implying that the solid should contain considerably amounts of nitrate. I presume the organisms can both use nitrate as electron acceptor or not? What would be the implications of the presence of nitrate.

Minor text related comments The first two sentences in the abstract do not help to grasp the content of the study but obscure the subject. My first impression was that the authors argue that isotopic fractionation is the cause for the cessation of iron reduction. Line 37, I presume the final rates were only a few percent of the initial rate (reformulate).

---

## Referee Comment (RC2) · Anonymous Referee #2 · 22 Jun 2020

The manuscript "Interpretation of kinetic isotopic fractionation between aqueous Fe(II) and ferrihydrite under a high degree of microbial reduction" compares Fe isotopic fractionation between Fe(II) and Fe(III) in ferrihydrite during ferryhydrite reduction by two strains of Shewanella at atmospheric pressure and 15 MPa.

General comments The main goal of the study and the research questions are not clearly stated. The use of pressure is not discussed. It is unclear what the pressure experiments bring to the study. In itself, the role of pressure on Fe isotope fractionation is a valid question (with an adequate experimental design), but it is only mentioned in the abstract and is apparently used in the study as a way to modulate Fe(III) reduction

rates. It would have been beneficial to determine the mineralogy of Fe minerals as a function of time. Fe(II) catalyzes mineral transformation at the surface of FeIII oxides and mineral transformation might potentially influence Fe fractionation.

Specific comments First sentence of abstract is unclear about what the topic of the study is. l.55-56 reference missing for the procedure (Schwertmann & Cornell) l.57-60 should be moved to "Iron isotope measurements section" l.71 It is unclear what the DIR experiment media are l.72 what is the concentration of ferrihydrite? What is the starting cell concentration? Figure 1: why is the total concentration of FeII decreasing at incubations of WP3 at ambient pressure? l.112-113 What does the ratio of FeII sorbed to FeII aqueous indicate? What is its significance? l. 131-135 The effects of pressure on Fe(III) reduction have been previously investigated for S. piezotolerans WP3 (Wu et al. 2013 Geobiology) and for Shewanella profunda (Picard et al. 2015 Frontiers). l.137-139 not all appropriate references are used

Technical comments l.31 "forming a wide range of soluble Fe(II)": replace by "producing soluble Fe(II)" l.35 replace ", in showing" by "have shown" l.36-37 Be specific: are you talking about the yield of the reaction (how much Fe(III) is reduced overall during the experiment) or the rate at a specific time l.38 Replace "Seems to no effect on" by "does not seem to impact" l.46 Replace "less" by "low" l.61 Plural of medium is "Media" Figure 1: typo in piezotolerans in the figure panels a and c. Also found throughout the manuscript l.102 and after: mM instead of mM L-1 l.109-110 It is a well-established fact that low crystalline Fe(III) minerals are reduced more and faster than crystalline Fe(III) oxides. Use appropriate references

---

## Author Comment (AC1) · 11 Aug 2020

We sincerely thank Reviewer 1 to provide feedback on our manuscript.

1. As the authors realize, ferrihydrite readily transforms into secondary minerals in the presence of Fe(II) depending on, among other factors, Fe(II) concentration. Hence, it can be assumed that different types of secondary iron minerals have been formed in the experiments depending on the rates and extent of Fe(III) reduction. Changes in mineralogy, obviously, effect isotope fractionation and without quantitative information of the Fe isotope signature of the various Fe species it is very difficult to interpret fractionation factor.

[Figure]

Author response: Thank you for your suggestion, it would have been benificial to determine the mineralogy of Fe minerals and Fe isotope fractionation as a function of time. We surely haven't done more research about this area except for SEM/TEM study. Previous studies suggested that biogenic magnetite retained morphologic and size features of ferrihydrite, wheareas siderite (another readily formed secondary mineral during ferrihydrite bioreduction) was generally observed as rhombohedral crystallites (Zachara et al., 2002). Determined by SEM, the reduction end-product in our experiment surely produced a little of magnetite. This is consistent with the research of Wu et al., (2013), whose materials and methods we refered to. When magnetite is the only secondary mineral in the HFO reduction experiment, Fe isotope fractionation is mainly associated with HFO reduction rate (Johnson et al., 2005). In a long-term laboratory experiment at low Fe(III) reduction rate, the Fe(II)aq-magnetite fractionation have been achieved at constant of -1.3‰ which is interpreted to be the equilibrium fractionation factor at 22°C. However, at a high ferrihydrite reduction rate experiment, Fe isotope fractionation between Fe(II)aq and ferrihydrite substrate is essentially associated with rapid sorption of Fe(II) to HFO. Moreover, magnetite is usually produced at the second of half of the experiment. As our experiments performed at a more higher reduction rate than Jonhnson'sïijŇand the reaction is short, so the influence of magnetite on Fe isotope fraction was limited. We will state it more clearly in the revised version.

2. I also have several other concerns about the interpretation of the data: According to the methodology about 1 g Fh were added to 50 mL medium. This should yield a Fe concentration of about 120 mM. This implies that only around 50 % of total Fe was recovered, which questions the isotope values for Fe(III) when the digestion was not quantitative. The trend that Fe recovery decreases with progressing reaction might reflect Fe mineral transformation (e.g. magnetite formation).

Author response: We are sorry for making a clerical error. The amount of ferrihydrite added to the 50 mL medium was 0.1 g, which should yield a final Fe concentration of about 20 ∼ 22 mM according to the controversial ferrihydrite chemical formula of

5Fe2O3.9H2O or Fe5HO8.4H2O. The concentrations of Fe0.1 M HCl and Fe0.5 M HCl in Table 1 represent the extracts (not the reactor), and the volumes of 0.1 M HCl and 0.5 M HCl extracts used for extraction are 15 ml and 20ml, respectively. However, the concentration of Feaq in Table 1 represents the reactor of which the volume is 50 ml. So, this leads to the misunderstanding that the recovery is only around 50%. We will correct it and state clearly in the revised version. 3. The ratio Fe(II)(0.1M HCl)/(Fetot(0.1M HCl) +Fetot(0.5M HCl)) exceeds 0.25, which is larger than a realistic concentration of surface sites (about 0.2 per Fe for HFO). This implies that not all extracted Fe(II) is adsorbed Fe but includes structurally bound Fe(II).

Author response: The ratio doesn't exceed realistic concentration of surface sites, the detail reason see response 2. We will state it clearly in the manuscript.

4. The authors do not mention anything about pH. Does the pH change throughout the reaction (no buffer is present in the medium) and how would pH effect fractionation. Considering these uncertainties, I am sceptic that the data set could be used to rigorously discussing fractionation mechanisms or deriving reliable fractionation factors.

Author response: Our apologies for no mention about pH and the effect of its variation on the rate and extent of Fe isotope exchange in the manuscript. In fact, the pH of each aqueous fraction was determined by HQ 40d in our experiments. The initial value of S. piezotolerans WP3 and S. oneidensis MR-1 reduction experiments were 6.3 and 6.6, respectively. With the proceeding of reaction, the pH increased to 6.4~6.8 in S. piezotolerans WP3 reactor, as well as 6.8~7.3 in S. oneidensis MR-1 reactor. The effect of pH on Fe isotope fractionation is essentially attributed to that Fe(II) sorption onto ferric minerals (Reddy et al., 2015). We will add the pH section and disscuss its influence on Fe isotope fractionation in the revised version.

5. I have also a couple of minor comments: Why did the authors vary the pressure? The experimental design is not justified. Varying the reduction rates or manipulating the Fe(II)/Fe(tot) ratios could have been easier achieved by adapting the bacteria/Fh

ratio. Using different organisms and pressures creates unnecessary ambiguity.

Author response: We agree with your comments. We chosen pressure and bacterial strains as a way to modulate reduction rates and extents. The results haven show that the effect of pressure on the extent of bioreduction and Fe isotope fractionation is not obvious. However, the bacterial strains have significantly impact on the rate and extent of bioreduction, and Fe isotope fractionation. In order to clarify the fact that Fe isotope exchange will be inhibited under high degree of bioreduction and the comments you give, we will remove the pressure part in the revised version.

6. Fh is produced by neutralizing a Fe(III)NO3 solution with KOH. The authors do not mention any purification step before freeze drying, implying that the solid should contain considerably amounts of nitrate. I presume the organisms can both use nitrate as electron acceptor or not? What would be the implications of the presence of nitrate.

Author response: Our apologies for no mention about the purification steps. Before freeze drying, ultrapure water was added to the suspension and centrifuged to isolate the nitrate fraction, repeating this operation 10 times. We will add this part in the revised manuscript.

7. Minor text related comments: The first two sentences in the abstract do not help to grasp the content of the study but obscure the subject. My first impression was that the authors argue that isotopic fractionation is the cause for the cessation of iron reduction.

Author response: We agree with your comments, and we will remove the two sentences in the reviesed version.

References:

Johnson, C. M., Roden, E. E., Welch, S. A., and Beard, B. L.: Experimental constraints on Fe isotope fractionation during magnetite and Fe carbonate formation coupled to dissimilatory hydrous ferric oxide reduction, Geochimica et Cosmochimica Acta, 69, 963-993, https://doi.org/10.1016/j.gca.2004.06.043, 2005.

[Figure]

Reddy, T. R., Frierdich, A. J., Beard, B. L., and Johnson, C. M.: The effect of pH on stable iron isotope exchange and fractionation between aqueous Fe(II) and goethite, Chemical Geology, 397, 118-127, https://doi.org/10.1016/j.chemgeo.2015.01.018, 2015.

Wu, W. F., Wang, F. P., Li, J. H., Yang, X. W., Xiao, X., and Pan, Y. X.: Iron reduction and mineralization of deep-sea iron reducing bacterium Shewanella piezotolerans WP3 at elevated hydrostatic pressures, Geobiology, 11, 593-601, https://doi.org/10.1111/gbi.12061, 2013.

Zachara, J. M., Kukkadapu, R. K., Fredrickson, J. K., Gorby, Y. A., and Smith, S. C.: Biomineralization of poorly crystalline Fe(III) oxides by dissimilatory metal reducing bacteria (DMRB), Geomicrobiology Journal, 19, 179-207, https://doi.org/10.1080/01490450252864271, 2002.

---

## Author Comment (AC2) · 11 Aug 2020

We thank the Anonymous Referee #2 for his comprehensive comments, we will respond to each below.

General comments:

1. The main goal of the study and the research questions are not clearly stated. The use of pressure is not discussed. It is unclear what the pressure experiments bring to the study. In itself, the role of pressure on Fe isotope fractionation is a valid question (with an adequate experimental design), but it is only mentioned in the abstract and is

apparently used in the study as a way to modulate Fe(III) reduction rate.

Authors response: In order to obtain the different extents of bioreduction and its relationship with Fe isotope fracitonation, we chosen pressure and bacterial strains as a way to modulate it. The results show that the effect of pressure on the extent of bioreduction and Fe isotope fractionation is not obvious. However, the bacterial strains have significantly impact on the rate and extent of bioreduction, and Fe isotope fractionation will be impacted under higher degree of bioredcution. In order to clarify the fact that Fe isotope exchange will be inhibited under high degree of bioreduction, we will remove the pressure section in the revised paper.

2. It would have been beneficial to determine the mineralogy of Fe minerals as a function of time. Fe(II) catalyzes mineral transformation at the surface of Fe(âĚ́) oxides and mineral transformation might potentially influence Fe fractionation.

Authors response: Please see the response 1 to Anonymous Referee #1.

Specific comments:

3. First sentence of abstract is unclear about what the topic of the study is.

Author response: We agree with your comments, it surely obscures the subject. We will write it clearly in the revised manuscript.

4. l.55-56 reference missing for the procedure (Schwertmann & Cornell)

Author response: Thank you for your suggestion. It will be done.

5. l.57-60 should be moved to "Iron isotope measurements section".

Author response: We agree with your comment. We will move it to "Fe isotope measurements section".

6. l.71 It is unclear what the DIR experiment media are

Author response: We will clarify it in the manuscript. 50 ml 2216E and LB media were

added into sterile plastic syringes, respectively, followed by addition of 0.1 g ferrihydrite and bacteria. The syringes were then sealed with PE material stoppers and placed in the high pressure steel vessel for the DIR experiments.

7. l.72 what is the concentration of ferrihydrite? What is the starting cell concentration?

Author response: We add 0.1g ferrihydrite to the 50 ml medium. It should yield a concentration of about $20 \sim 22$ mM according to the controversial ferrihydrite chemical formula of 5Fe2O3.9H2O or Fe5HO8.4H2O. The starting cell concentration was about $7.22 \times 10$-6 L-1. We will state it clearly in the manuscript.

8. Figure 1: why is the total concentration of Fe(II) decreasing at incubations of WP3 at ambient pressure?

Author response: We have no idea about this. It may be due to the formation of secondary minerals or experimental operation error.

9. l.112-113 What does the ratio of Fe(II) sorbed to Fe(III) aqueous indicate? What is its significance?

Author response: Thank you for your suggestion. This is a problematic statement, we will remove these sentences in the revised version.

10. l. 131-135 The eeffects of pressure on Fe(III) reduction have been previously investigated for S. piezotolerans WP3 (Wu et al. 2013 Geobiology) and for Shewanella profunda (Picard et al. 2015 Frontiers). l.137-139 not all appropriate references are used.

Author response: We will clarify it.

Technical comments:

11. l.31"forming a wide range of soluble Fe(II)": replace by "producing soluble Fe(II)" l.35 replace ", in showing" by "have shown" l.38 Replace "Seems to no effect on " by "does not seem to impact" l.46 Replace "less" by "low" l.61 Plural of medium is "Media"

Figure 1: typo in piezotolerans in the figure panels a and c. Also found throughout the manuscript. l.102 and after: mM instead of mM L-1

Author response: We thank you for pointing out the errors in the manuscript. We will amend these errors in the manuscript.

12. l.36-37 Be specific: are you talking about the yield of the reaction (how much Fe(III) is reduced overall during the experiment) or the rate at a specific time.

Author response: Yes, we want to show the degree of hematite and goethite bioreduction is lower than 4% in a lasting 280 days experiment. But, we made a clerical error. The "rate" in l.37 shoud be replaced by "extent".

13. l.109-110 It is a well-established fact that low crystalline Fe(III) minerals are reduced more and faster than crystalline Fe(III) oxides. Use appropriate references

Author response: Yes, this is also the reason that we choose ferrihydrite as terminal ferric substance to perform DIR experiments. We will cite the appropriate references in the revised manuscript.